# Myogenesis in C2C12 Cells Requires Phosphorylation of ATF6α by p38 MAPK

**DOI:** 10.3390/biomedicines11051457

**Published:** 2023-05-16

**Authors:** Valentina Pagliara, Giuseppina Amodio, Vincenzo Vestuto, Silvia Franceschelli, Nicola Antonino Russo, Vittorio Cirillo, Giovanna Mottola, Paolo Remondelli, Ornella Moltedo

**Affiliations:** 1Department of Medicine, Surgery and Dentistry “Scuola Medica Salernitana”, University of Salerno, Via Salvador Allende, 84081 Baronissi, Italy; vpagliara@unisa.it (V.P.); gamodio@unisa.it (G.A.); 2Department of Pharmacy, University of Salerno, Via Giovanni Paolo II, 84084 Fisciano, Italy; vvestuto@unisa.it (V.V.); sfranceschelli@unisa.it (S.F.); v.cirillo19@studenti.unisa.it (V.C.); 3Biogem, Istituto di Biologia e Genetica Molecolare, Via Camporeale, 83031 Ariano Irpino, Italy; nicola.russo@biogem.it; 4Centre de Recherche en Cardiovasculaire et Nutrition (C2VN) (AMU-INSERM 1263-INRAE 1260), Aix Marseille Université, Campus Timone, 27 Bd. Jean Moulin, 13005 Marseille, France; giovanna.mottola@univ-amu.fr; 5Biogénopôle (BGP), Laboratoires de Biologie Médicale, Secteur Biochimie, Hôpital de La Timone, 264 Rue Saint-Pierre, 13005 Marseille, France

**Keywords:** C2C12, myogenesis, unfolded protein response, activating transcription factor 6 α (ATF6α), p38 Mitogen-Activated Protein Kinase (MAPK)

## Abstract

Activating transcription factor 6α (ATF6α) is an endoplasmic reticulum protein known to participate in unfolded protein response (UPR) during ER stress in mammals. Herein, we show that in mouse C2C12 myoblasts induced to differentiate, ATF6α is the only pathway of the UPR activated. ATF6α stimulation is p38 MAPK-dependent, as revealed by the use of the inhibitor SB203580, which halts myotube formation and, at the same time, impairs trafficking of ATF6α, which accumulates at the cis-Golgi without being processed in the p50 transcriptional active form. To further evaluate the role of ATF6α, we knocked out the ATF6α gene, thus inhibiting the C2C12 myoblast from undergoing myogenesis, and this occurred independently from p38 MAPK activity. The expression of exogenous ATF6α in knocked-out ATF6α cells recover myogenesis, whereas the expression of an ATF6α mutant in the p38 MAPK phosphorylation site (T166) was not able to regain myogenesis. Genetic ablation of ATF6α also prevents the exit from the cell cycle, which is essential for muscle differentiation. Furthermore, when we inhibited differentiation by the use of dexamethasone in C2C12 cells, we found inactivation of p38 MAPK and, consequently, loss of ATF6α activity. All these findings suggest that the p-p38 MAPK/ATF6α axis, in pathophysiological conditions, regulates myogenesis by promoting the exit from the cell cycle, an essential step to start myoblasts differentiation.

## 1. Introduction

In differentiated skeletal muscle cells, the endoplasmic reticulum (ER) membranes expand like a continuous network, which is called sarcoplasmic reticulum (SR). Here, specialized domains are devoted to the control of Ca^++^ storage and release required for muscle contraction [1,2]. Apart from this, ER membranes house the entire signal transduction battery of the unfolded protein response (UPR) consisting of the protein kinase RNA-like Endoplasmic Reticulum kinase (PERK), the Inositol-requiring protein 1 (IRE1) and the activating transcription factor 6α (ATF6α) [3,4,5]. The UPR is finely regulated by the glucose-regulated protein 78 (GRP78), also referred to as BiP, which under ER stress, binds misfolded proteins accumulating within the ER, leaving each UPR transducer-free [5,6]. In this way, these proteins can activate distinct pathways to coordinate a defensive response, aiming to reduce the engulfment of misfolded proteins within the ER. This condition is known as ER stress, against which the UPR responds by activating the transcription and/or translation of folding factors operating the quality control (QC) of protein folding within the ER [7,8,9]. In response to ER stress, the UPR reduces the amount of protein entering the ER by downregulation protein synthesis and enhances misfolded proteins degradation by activating the ubiquitin/proteasomal and/or autophagy pathways [10,11,12,13]. Indeed, ER stress is usually a transitory disorder, but when the stress is irreversible, the strong imbalance of the redox homeostasis and the accumulation of toxic aggregates initiate the cell death program, which is in part promoted by the UPR itself [14].

Myogenesis is a multistep process, in which myoblasts arrest cell division and then start to elongate and fuse forming multinucleated myotubes [15]. These events take place in both muscle development and in muscle regeneration [16] and are driven by the expression of myogenic regulatory factors (MRFs) including myogenic factor 5 (Myf5) [17], the myoblast determination protein 1 (MyoD) [18], myogenin (MyoG) [19], the myogenic regulatory factor 4 (MRF4) [20] and, finally, the myocyte enhancer factor (MEF2) [21].

A number of investigations indicate that the p38 MAPK signaling presides over myogenic differentiation or cross-talks with the UPR [22,23]. Indeed, inhibition of the p38 MAPK blocks expression of muscle-specific genes and myotube formation [24]. 

Although ER stress activates simultaneous with the UPR pathways [10,14], differential regulation of the single pathways occurs in several physiological processes. For example, the thyroglobulin synthesis in thyrocytes activates PERK and ATF6α but not the splicing of XBP-1 mRNA by IRE1 [25]. Similar to what we observe in C2C12 myogenesis, differentiating B cells undergo an extensive expansion of the ER network and an increased expression of ER chaperones, which allows cells to produce huge amounts of Ig [26]. In a similar way to what we see, this kind of response demands ATF6α activation, whereas PERK is specifically suppressed to warrant normal protein synthesis, which is highly necessary during myotube development [16,27,28].

An inverse relationship between the PERK activity and muscle differentiation is quite common. Indeed, an increased expression of CHOP, a downstream target of PERK, inhibits myogenic differentiation by repressing the transcription factor MyoD [29]. In other examples, such as upon muscle injury, PERK is required to expand satellite cells (SCs) at the regeneration site, where SCs will differentiate [28,30,31]. Thus, PERK is constitutively active in quiescent SCs. However, when these cells start differentiation, both PERK and the phosphorylation of its downstream target, eIF2α, are reduced. Moreover, PERK ablation or the expression eIF2α phosphorylation mutant (eIF2αS51A) generates a loss of SCs’ quiescence state and drives the induction of differentiation [30].

Concerning IRE1, a number of reports suggest that this branch of the UPR could give opposite results in myoblasts differentiation. Indeed, while on the one hand both IRE1 and XBP1 knockdown suppress C2C12 myoblasts differentiation [32], XBP1 overexpression inhibits the expression of myogenic factors as well as the formation of myotubes by enhancing the expression of Mist1, a negative regulator of MyoD [33]. Furthermore, although chronic stress can induce IRE1 to activate pro-apoptotic c-Jun N-terminal kinase (JNK) [34], during differentiation, the JNK/MAPK pathway is downregulated, and IRE1α inhibition leads to the hyperactivation of p38 MAPK (but not of JNK) and myotubes formation in C2C12 cells [35,36]. Therefore, as regards IRE1, our results support the idea that this pathway of the UPR is not necessary for myoblasts differentiation. 

To date, few studies have explored the role of ATF6α in promoting myogenesis, and little is known about the involvement of ATF6α in the development of skeletal muscles [27,37,38]. In other contexts, p38 MAPK phosphorylates a threonine residue at position 166 of the ATF6α protein, and mutants at this site fail to undergo proteolytic processing at the Golgi complex, indicating that ATF6α phosphorylation by p38 MAPK is crucial for the proteolytic cleavage, which transforms ATF6α into the transcription factor p50 ATF6α [39,40]. Therefore, we investigated the activation modes and the effect of the genetic ablation of ATF6α on the myogenesis of mouse myoblast cell line C2C12. Our results suggest, for the first time, the important role of the p38 MAPK/ATF6α pathway during muscle cell differentiation.

## 2. Materials and Methods

### 2.1. Cell Culture and Treatments

Mouse C2C12 myoblasts were purchased from the American Type Culture Collection (ATCC, Rockville, MD, USA) [41]. Cells were cultured in a growth medium (GM) containing Dulbecco’s Modified Eagle’s Medium (DMEM, Gibco, Invitrogen, Grand Island, NY, USA) supplemented with a 10% fetal bovine serum (FBS) and 1% penicillin/streptomycin at 37 °C and 5% CO_2_. Myotube formation was induced by growing C2C12 myoblasts in a differentiation medium (DM) containing 2% horse serum and 1% penicillin/streptomycin for 24, 48 and 72 h. Cells were incubated for 5 h with tunicamycin (TN) (Sigma–Aldrich, Darmstadt, Germany) 2 μg/mL to induce ER stress [41]. Treatment with 10 μM p38 inhibitor SB203580 (SB) (Sigma–Aldrich, Darmstadt, Germany) was performed for 72 h during myotube induction in DM [42]. For Dexamethasone (DEX) (Sigma–Aldrich, Darmstadt, Germany), treatment cells were induced in DM for 24 h and then incubated in the same medium for 48 h with 10 μM DEX.

### 2.2. Genetic Ablation of the ATF6α Gene and Isolation of Knock out ATF6α C2C12 Myoblasts

*ATF6α* knockout was performed using the lentiCRISPRv2 plasmid to knockout *ATF6α*. Short guide RNAs (sg RNAs) against ATF6α [43] 0.1 pmoL/μL sg RNA pairs (5-TTTAGTCCGGTTCTTCCTCAT-3 and 5-ATGAGGAAGAACCGGACTAAA-3) was introduced to the lentiCRISPRv2 plasmid through digestion/ligation cycles using BsmBI and T4 ligase (NEBridge^®^ Golden Gate Assembly Kit E1602S, New England Biolabs, Ipswich, MA, USA). Stable transfectants were selected in a growth medium containing 4 μg/mL Puromycin (Sigma–Aldrich, Darmstadt, Germany). Sub-confluent cells were transfected 24 h after plating by using TRANSIT (Mirus, Madison, WI, USA). Empty lentiCRISPRv2 vectors were used to generate clones as an un-edited control (EV), while un-transfected C2C12 cells were used as a basal control. Stable transfectants were isolated by collecting individual clones, which were further amplified. Single clones and bulk culture (pool) were analyzed by Western blotting to verify gene inactivation. Clone 8 was chosen for further characterization (Appendix A).

### 2.3. Construction of T166A and S130A ATF6α Mutants

Plasmid expressing p3xFlagATF6α (Addgene, Watertown, MA, USA) was used as a template for mutagenesis to obtain mutants T166A [44] and S130A, identified as putative p38 MAPK phosphorylation sites in ATF6α by using the NetPhos 3.1 Server (https://services.healthtech.dtu.dk/service.php?NetPhos-3.1, accessed on 15 September 2022), which revealed seven putative p38 MAPK phosphorylation sites in the ATF6 protein sequence: S13; S16; S94; S130; T166; and S548. Phosphorylation sites T166 and S130, not yet characterized, were chosen for the experimental procedures. Mutagenesis of plasmid p3xFlag-ATF6α, expressing full-length wild type human ATF6α (Addgen, MA, USA), was obtained by using the In-Fusion Cloning Kit (Takara Bio, Shiga, Japan). Two different pairs of Inverse primers (FW-S130A 5′-TCAGATGGCCCCCCTTTCCTTATATGGTGAAAAC-3′ and Rev-S130A 5′-AGGGGGGCCATCTGAGAACTAGAAGACAAATCC-3′; FW-T166A 5′-TGGACTGGCCCCAAAGAAAAAAATTCAGGTGAA-3′ and Rev-T166A TTTGGGGCCAGTCCATTTTCAGTCTTGTTCCT) were used to amplify the p3xFlag-ATF6α vector and to obtain serine/alanine (S130A) and threonine/alanine (T166A) substitution, respectively. For the In-Fusion Cloning reaction, the linear DNA was re-circularized at the site of the 15 bp overlap and mutagenic changes were included. Finally, constructs were validated by Sanger sequencing (Eurofins Genomics Services, Ebersberg, Germany).

### 2.4. Western Blotting Analysis

Whole-cell lysate was prepared by lysing cells in a solution containing 50 mM Tris–HCl pH 8.0, 150 mM NaCl, 0.5% sodium deoxycholate, 0.1% SDS, 1 mM EDTA, 1% Igepal, 1× protease inhibitor and a phosphatase inhibitor cocktail, as described previously [45,46]. Protein concentration was determined by the Bradford protein assay. Proteins were loaded on 10% SDS–PAGE and detected by Western blotting. After electrophoresis, proteins were transferred to a nitrocellulose membrane and then incubated with the specific primary antibody. Prestained Protein Ladders were purchased from Bio-rad laboratories, Inc. Italy. The following antibodies were used: mouse monoclonal antibody raised against MyoG; mouse monoclonal antibody raised against p-p38 MAPK; mouse polyclonal antibody raised against p38; mouse monoclonal antibody raised against GAPDH; rabbit monoclonal antibody raised against ATF6 (Santa Cruz Biotechnology, Dallas, TX, USA); rabbit monoclonal antibody raised against Calnexin (Cell Signaling, Danvers, MA, USA); and mouse monoclonal antibody raised against GM130 (BD Transduction Laboratory, San Jose, CA, USA). After incubation with the appropriate anti-rabbit or anti-mouse (Pierce, Thermo Fisher Scientific, Waltham, MA, USA) peroxidase-linked secondary antibody, detection was achieved using the Enhanced Chemiluminescence (ECL) kit (Advansta, San Jose, CA, USA). Densitometry analysis was performed using the free image-processing software ImageJ version 1.47 (http://rsb.info.nih.gov/ij/, accessed on 10 November 2022).

### 2.5. RNA Extraction, Reverse Transcription (RT), XBPI Splicing Assay and Quantitative Real-Time Polymerase Chain Reaction (qPCR)

Total RNA was purified by using an ultrapure TRizol reagent (Gibco, Thermo Fisher Scientific, Whaltam, MA, USA) according to the manufacturer’s instructions. The concentration and purity of RNA were determined spectrophotometrically by reading the absorbance at 260 and 280 nm. Aliquots of total RNA were subjected to DNase I digestion (Thermo Fisher Scientific, Waltham, MA, USA) and reverse transcribed using EasyScript Plus cDNA Synthesis Kit (abm, Vancouver, BC, Canada) according to the manufacturer’s protocol. Real-time PCR was carried out using the PowerUP Syber green master mix (Thermo Fisher Scientific, MA, USA), the Quant Studio 7 Flex instrument and the fast gene-expression method with the following conditions: a first denaturation step at 95 °C for 20 s followed by 40 cycles at 95 °C for 1 s, 60 °C for 30 s, and then melting curve analysis was performed, raising the temperature from 60 °C until 95 °C with a 0.5 °C/s increase. Reactions were carried out in triplicate, and the 18S gene was used as an internal control to normalize the variability in expression levels. The 2^−ΔΔCT^ (cycle threshold) method was used to calculate the results, and mRNA expression levels were determined as fold-induction relative to the Ctrl cells, set as 1 [47]. The primers used for the qPCR reactions are the following: GRP78/BiP forward, 5′-TCTGGTGATCAGGATACAGGTG-3′; GRP78/BiP reverse, 5′ATGATTGTCTTTTGTTAGGGGTGC-3′; Calnexin forward, 5′-GGTCTCTGTCAGGGTGGATTTTAT-3′; Calnexin reverse, 5′-TGGAAGCTTTGTTTCCTTCATCTC-3′; Calreticulin forward, 5′-CCTGAATACTCCCCCGATGC-3′; Calreticulin reverse, 5′-ATTGTCCCGGACTTGACCTG-3′; MyoG forward, 5′-AGGAGATTTGCTCGCGG-3′; MyoG reverse, 5′-CAGTTGGGCATGGTTTCGTC-3′; 18S forward, 5′-CGGCTACCACATCCAAGGAA-3′; and 18S reverse, 5′-GGGCCTCGAAAGAGTCCTGT-3′. Primers were designed using NCBI primer Blast. NM_ accession of each gene was retrieved from https://www.ncbi.nlm.nih.gov/nucleotide (accessed on 4 October 2021) and used as input, in order to use the mRNA sequence for primer design. The primer melting temperature was set to be 60 °C. The “Exon junction span” parameter was selected as “primer must span an exon-exon junction” and “Intron inclusion” parameter was flagged. Finally, “Primer specificity stringency” was set as “Primer must have at least 3 total mismatches to unintended targets, including at least 3 mismatches within the last 5 bps at the 3′ end”.

RT-PCR and XBPI Splicing Assay: Total RNA was retro-transcribed with the EasyScript Plus cDNA Synthesis Kit (abm, Vancouver, BC, Canada) according to manufacturer instructions. To amplify the mouse XBP1 mRNA, PCR was carried out for 30 cycles (94 °C for 2 min; 55 °C for 30 s; and 72 °C for 2 min followed by 10 min at 72 °C) using the following primers forward, 5′-CCTTGTGGTTGAGAACCACC-3′; and reverse, 5′-CTAGAGGCTTGGTGTATA-3′, as previously described [48]. Un-spliced and spliced XBP1 mRNA were separated by gel electrophoresis on 3% agarose gel. Safeview (abm, Vancouver, Canada)-stained amplicons were quantified by densitometry with ImageJ version 1.47 (http://rsb.info.nih.gov/ij/, accessed on 10 November 2022).

### 2.6. Phase-Contrast and Confocal Microscopy

To perform immunofluorescence analysis, undifferentiated and differentiated cells were washed in phosphate-buffered saline (PBS), fixed in PBS-4% paraformaldehyde and permeabilized with 0.1% Triton X-100 in PBS for 5 min [49]. Thereafter, cells were stained with Phalloidin-Tetra-methyl-rhodamine B isothiocyanate (Merk, Darmstadt, Germany); antibodies anti-GRASP 65 or anti-GM130 was used to visualize the Golgi (BD Transduction Laboratory, San Jose, CA, USA) for 1 h; rabbit antibody was raised against ATF6 (Santa Cruz Biotechnology, Dallas, TX, USA); to visualize ER in vivo staining was performed by using an endoplasmic reticulum-tracker (ER-tracker) (Invitrogen, Whaltam, MA, USA); rabbit monoclonal anti-Flag antibody was used to visualize 3XFLAG ATF6α (Cell Signaling, Danvers, MA, USA) for 1 h; and 4′,6-diamidino-2-phenylindole (DAPI) was used to visualize the nuclei. After washing, coverslips were mounted with a Vecta-mount medium. Images were acquired with a laser scanning confocal microscope TCS SP5 (Leica MicroSystems, Mannheim, Germany) equipped with a plan Apo 40X, NA 1.4 oil immersion objective lens. Pictures were processed using LAS-AF Software (Leica MicroSystems, Germany) to reconstruct the *x*-axis projection using stack images [50]. Co-localization analysis was performed by using Leica SP5 of 30 cells for each sample, as previously described [51]. The co-localization rate reported in figures was calculated by using the proprietary co-localization algorithm in Leica Software (LAS-AF 2.7.3.9723). The value indicates the extent of co-localization as a percentage and is calculated for each pair of fluorophores from the ratio between the number of co-localizing fluorescent pixels and the number of the total fluorescent pixels of the two fluorophores in the image. In particular, the co-localization rate of ATF6 vs. DAPI was calculated by dividing their co-localizing pixels by the sum of ATF6 and DAPI pixels (number of ATF6-DAPI co-localizing pixels/number of ATF6 + DAPI pixels). The same algorithm was used to calculate the rate of ATF6-Golgi co-localization (number of ATF6-Golgi co-localizing pixels/number of ATF6 + Golgi pixels) and ATF6-ER co-localization (number of ATF6-ER co-localizing pixels/number of ATF6 + ER pixels). Phase-contrast images were captured using a Leica DM IL LED inverted microscope (10× objective) (Meyer Instruments, Huston, TX, USA).

### 2.7. Evaluation of Apoptosis and Cell Cycle

To determine the number of apoptotic nuclei and cell cycle phases analysis, cells (4 × 10^5^ cells/well) were seeded into 12-well plates; at the end of each treatment, cell suspensions were centrifuged, and pellets were resuspended in a hypotonic lysis solution containing 50 µg/mL propidium iodide (PI). After incubation at 4 °C for 30 min, cells were analyzed by a Becton Dickinson FACScan flow cytometer, using the Cell Quest software version 4 [52,53]. Cellular debris was excluded from the analysis by raising the forward scatter threshold, then the percentage of cells in the hypodiploid region (sub G0/G1), G1, G2 and S phase were calculated.

### 2.8. Statistical Analysis

Statistical significance was determined by one-way analysis of variance (ANOVA), followed by Bonferroni test. Each value represents the mean ± SD of at least three independent experiments performed in triplicate (* *p* < 0.05, ** *p* < 0.01, *** *p* < 0.001).

## 3. Results

### 3.1. ATF6α Is the Only UPR Pathway Activated in Differentiating Mouse C2C12 Myoblasts

First, we tested whether myogenesis of C2C12 myoblasts requires the unfolded protein response by analyzing the activity of each component of the UPR pathways. 

The progress of myogenesis was assessed by culturing cells in a differentiating medium (DM) supplemented with 2% horse serum (Figure 1). 

Phase-contrast microscopy revealed that the formation of myotubes already appeared at 24 h, following cell incubation in DM. However, the higher number of multinucleated cells was obtained after 72 h in DM compared to the control cells kept in a normal growth medium (GM) (Figure 1A). The morphological results were confirmed by the increased levels at the different time points of C2C12 differentiation of MyoG mRNA (Figure 1B) as well as of the MyoG protein (Figure 1C), as determined by quantitative real-time RT-PCR and Western blotting, respectively.

Activation of the endogenous ATF6α was assessed by measuring the amount of the un-cleaved p90-ATF6α protein against the cleaved p50 form of ATF6α in the C2C12 myoblast, cultured in GM or in DM for the times indicated (Figure 1D). As such, in the cells grown in GM we observed, by Western blotting, only the un-cleaved p90-kDa form of ATF6α, proving that the UPR transducer is not active in the undifferentiating myoblasts. Instead, under differentiation, we revealed that ATF6α was in the p50 kDa form, which was increased by time, as shown by densitometric analysis in Figure 1E, demonstrating that ATF6α is activated during myogenesis. 

Interestingly, in our experiments, we did not observe an activation of either PERK or IRE1 (Figure 1F,G). To establish PERK activation, we analyzed, by Western blotting, the level of phosphorylated PERK form (p-PERK) expressed in C2C12 incubated in GM or DM for different time intervals (Figure 1F). PERK phosphorylation could be detected by the appearance of a band-shift of the PERK protein as a consequence of the higher molecular weight acquired by its auto-phosphorylation [48,54]. As we expected, results revealed that in the control cells exposed to tunicamycin (TN), we did detect the phosphorylated (p-PERK) form of the protein. Instead, we revealed that levels of unphosphorylated PERK were progressively reduced by DM incubation.

The IRE1 endonuclease activity was established by assaying the expression of the spliced form of XBP1 in the C2C12 mRNAs extracted at the different times of myogenesis (Figure 1G). Results showed that spliced XBP1 mRNA (426 bps) appeared only when the cells were treated with tunicamycin (TN), while all the other samples showed exclusively the un-spliced form (452 bps). As for PERK, we observed that the expression of the endogenous protein was even suppressed during the induction of C2C12 myogenesis, demonstrating that myotube biogenesis in differentiating C2C12 myoblasts does not necessitate both IRE1 and PERK activity. 

Finally, when we measured, by qPCR, the mRNA level of three markers of the ER stress, namely GRP78/BiP, Calnexin and Calreticulin (Figure 1H), we found that all of them were significantly increased (*p* ≤ 0.001) in differentiating myoblasts with respect to the cells growing in GM, suggesting that C2C12 myoblasts, during myogenesis, are sustained by ATF6α. 

### 3.2. p38 MAPK Phosphorylation Drives Activation of ATF6α during Myogenesis

Since it has long been known that the p38 MAPK signaling pathway drives myogenesis [55], we investigated whether the p38 MAPK is required for ATF6α activation. To this aim, C2C12 myoblasts were cultured in GM or DM for 72 h and incubated or not with the p38 MAPK inhibitor SB203580 (Figure 2). Western blot analysis (Figure 2A) revealed that SB203580 was able to reduce the cleavage of the endogenous ATF6α compared to what was found in the C2C12 myoblasts stimulated to differentiate, as also shown by the densitometric analysis (Figure 2B).

Similarly, we observed that either level of phosphorylated p38 MAPK (p-p38 MAPK) or MyoG expression were inhibited by the SB203580 at both the protein (Figure 2A) and mRNA (Figure 2D) levels in differentiating cells, confirming that p38 MAPK drives the expression of myogenic factors. Furthermore, as a consequence of the reduced ATF6α activation, we observed that, under differentiation, SB203580 consistently suppressed GRP78/BiP, Calnexin and Calreticulin mRNA expression (Figure 2C).

### 3.3. Inhibition of p38 MAPK by SB203580 Influences Morphology of the Secretory Pathway and the Intracellular Trafficking of the Endogenous ATF6α Protein

Thereafter, we investigated whether the inhibition of p38 MAPK by SB203580 could influence the endogenous ATF6α protein trafficking from the ER to the nucleus, via the Golgi complex, during C2C12 myoblasts differentiation. To this aim, C2C12 cells were cultured in GM or DM in the presence or absence of SB203580 and then subjected to confocal immune fluorescence analysis (Figure 3A). Co-localization analyses revealed that, in the myoblasts maintained in GM, endogenous ATF6α was mainly localized within the ER network stained by the ER-tracker (Figure 3A), indicating that endogenous p90 ATF6α does not exit the ER in resting cells. Instead, in differentiating cells, we observed an intense expansion of the ER membrane network and a redistribution of the Golgi complex, which was found interspersed throughout the multinucleated myotubes (Figure 3A). In this context, co-localization analyses showed that endogenous ATF6α localized at the cis-Golgi as well as within nuclei (Figure 3A, panels e, f: compare ATF6α fluorescence to GRASP 65 or DAPI fluorescence), indicating that p90 ATF6α undergoes proteolytical cleavage at the Golgi complex to release the p50 form, which enters nuclei. More interestingly, treatment with SB203580 not only impairs ER expansion and myotube formation (panels g–i) (Figure 3A) but also causes a build-up of the ATF6α protein in the Golgi apparatus and, concomitantly, the failure of ATF6α to localize at the nuclei (as shown by the % of co-localization analysis in Figure 3B). These data were corroborated by subcellular fractionation experiments (Appendix A), confirming that p38 MAPK is critical for the activation and trafficking of ATF6α as well as for myotube formation. 

### 3.4. Genetic Ablation of ATF6α Impairs Myogenesis of C2C12 Myoblasts

To understand how ATF6α can be important for myotube formation, we analyzed the effect of *ATF6α* knockout (KO) (Appendix A) on the C2C12 myoblast induced to myogenesis. To this purpose, we checked if the ablation of *ATF6α* could influence the expression of myogenic factors such as MyoG and the achievement of a differentiated phenotype (Figure 4). As we would expect, in a selected clone (C8) and in a pool of KO *ATF6α* C2C12 myoblasts, we did not find either ATF6α or MyoG protein expression and any formation of myotubes (Figure 4A,B) against un-transfected or empty vector transfected (EV) control cells. More interestingly, in KO ATF6α cells, phosphorylated p38 MAPK (p-p38) is still detectable in DM-stimulated cells, which proves that ATF6α is essential to achieve the differentiation state in C2C12 myoblasts. 

Next, we tested whether normal features of myogenesis were recovered by expressing exogenous ATF6α in KO *ATF6α* cells kept in GM or DM for 72 h and transfected with the wild-type 3xFlag-ATF6α expressing vector (Figure 5). Un-transfected cells were used as a negative control for the immunofluorescence experiment (Appendix A).

As we would expect, expression of exogenous ATF6α, recognized by the anti-Flag antibody, restored ER expansion and multinucleated myotube formation (panels d–f) after 72 h of differentiation time compared to the EV clone (Figure 5A: panels a–c). Fluorescent images and the co-localization rate (Figure 5B) clearly revealed that recombinant ATF6α localized at the Golgi complex and within nuclei, indicating that the cleavage of p90 ATF6α at the Golgi level was taking place, in the similar way as the endogenous ATF6α protein (Figure 3A).

As we would expect, SB203580 kept myotube formation blocked (Figure 5A: panels g–i), while ATF6α accumulated at the Golgi complex and was not found within the nuclei, indicating that the cleavage of p90 ATF6α was prevented (Figure 5B). These data were confirmed by Western blotting analyses, revealing that SB203580 suppresses recombinant ATF6α processing and MyoG expression, either at the protein or at the mRNA level (Figure 5C). Similarly, GRP78/BiP, Calnexin and Calreticulin mRNAs were also reduced, as we would have expected by the loss of ATF6α activity (Appendix A). These results confirm the requirement of the p38 MAPK for the ATF6α activity and strengthen the importance of ATF6α for myotube formation.

### 3.5. Myogenesis of C2C12 Requires Phosphorylation by P38 MAPK at the T166 Site of ATF6α 

We next asked whether ATF6α requires p38 MAPK phosphorylation at a threonine residue (T166) to drive the myogenesis of C2C12 myoblasts. To this aim, we analyzed myotube formation in KO ATF6α C2C12 cells transfected with vectors expressing wild-type or mutant T166A ATF6α recombinant proteins or the recombinant S130A ATF6α protein (Figure 6A). Immunofluorescence analysis (Figure 6B) revealed that the T166A ATF6α recombinant protein failed to restore ER expansion and the formation of multinucleated cells in KO *ATF6α* cells, during cell differentiation. Moreover, T166A ATF6α showed a higher accumulation at the Golgi complex (Figure 6B, panel e: compare anti-Flag to GM130) and failed to enter nuclei (Figure 6B, panel d: compare anti-Flag to DAPI). Instead, unlike recombinant T166A-ATF6α, the expression in differentiated KO-*ATF6α* cells of either the wild-type ATF6α construct (Figure 6C) or the S130A-ATF6α mutant did not affect myotubes formation and, as shown by Western blotting analyses, those proteins were normally processed (Figure 7A), suggesting that the serine 130 of ATF6α is not a phosphorylation site for p38 MAPK. Indeed, the MyoG was detectable only in the wild-type ATF6α and S130A-ATF6α transfected cells, whereas inhibition of the MyoG protein expression was observed only in the T166A ATF6α transfected cells, compared with the ATF6α WT or S130A- ATF6α transfected cells (Figure 7A). Furthermore, we observed a similar inhibitory effect, in the T166A-ATF6α transfected cells, on the expression of MyoG mRNA during KO *ATF6α* cells differentiation, compared to KO *ATF6α* cells transfected with WT ATF6α (Figure 7B). 

Similar inhibition was revealed for GRP78/BiP, Calnexin and Calreticulin mRNAs in T166A ATF6α -transfected cells with respect to the KO *ATF6α* cells transfected with WT ATF6 (Figure 7C), confirming that the T166A ATF6α mutant protein acts as a transcriptional factor. These results clearly indicate p38 MAPK phosphorylation at the T166 site for either ATF6α cleavage and its import to the nucleus and that it is essential for myotube biogenesis.

### 3.6. ATF6α Is Necessary for Cell Cycle Control during Myoblast Differentiation

Since the arrest of cell cycle is required to promote differentiation into muscle fibers [16], we asked whether the ATF6α could affect the cell cycle of C2C12 myoblasts induced to differentiate. The cytometric analysis, after PI incorporation, showed (Figure 8A) that EV control cells, when induced to differentiate, were mostly at the G1 phase and showed lower levels of cells in S and G2 compared to those kept in GM, suggesting that endogenous ATF6α enabled exit from the cell cycle, thereby inhibiting the proliferation of most of the cells. Instead, KO *ATF6α* cells, stimulated by DM, showed % of cells in each phase comparable to that found in the KO *ATF6α* cells kept in GM, indicating that ablation of ATF6α prevents the arrest of the cell cycle, which is essential to start myogenesis. To confirm this, transfection with WT ATF6α in KO *ATF6α* cells increased the percentage of cells in G1 (Figure 8B). Moreover, since no effect was observed in the KO *ATF6α* cells expressing the T166A ATF6α recombinant proteins (Figure 8B) and a similar effect was revealed following incubation with SB203580 given to the KO *ATF6α* cells transfected with the WT ATF6α (Figure 8B), this indicates the arrest of cell proliferation, which supports myogenesis of C2C12 cells, depending on the ATF6α phosphorylation performed by p38 MAPK.

### 3.7. p38 MAPK/ATF6α Pathway Regulates Dexamethasone-Induced Myotube Inhibition

Next, we asked whether the p38 MAPK/ATF6α pathway was altered in C2C12 myoblasts when differentiation was inhibited by incubating cells with dexamethasone (DEX). As expected, DEX treatment kept cells in an undifferentiated state (Figure 9A), compared to cells kept in DM alone. By these experiments, we also found that, following treatment of differentiating myoblasts with DEX, ATF6α highly localized at the Golgi membranes (Figure 9A, panel e), and nuclear localization of ATF6α was strongly lowered compared to the cells kept in DM alone (Figure 9A, panel d). Consistently, by analyzing the rate of co-localization with the cis-Golgi membranes labelled by the anti-GRASP 65 antibody, we found an increase in ATF6α co-localization with the cis-Golgi marker: (*p* ≤ 0.05) in the differentiated C2C12 cells treated with DEX and a significant reduction in the co-localization of the ATF6α fluorescent signal with the nuclear marker (*p* ≤ 0.01), compared to un-treated cells (Figure 9B). In line with this, Western blot analyses revealed that the inhibition of differentiation due to DEX decreased MyoG expression, the p38 MAPK activity and ATF6α processing, compared to the differentiated un-treated cells (Figure 9C).

## 4. Discussion

In this study, we report that myotube formation in C2C12 myoblasts requires an unconventional UPR supported by ATF6α but not by IRE1 or PERK. This was shown by stimulating differentiation in C2C12 cells and observing that ATF6α is progressively processed, which suggested that this pathway of the UPR, to a certain extent, could play a role in supporting myogenesis. Following this observation, we set up a series of experiments in order to understand through which mechanism ATF6α takes place to the control of myogenesis. 

The steps of ATF6a activation have been mainly studied in the UPR. Mammals express two homologous ATF6 proteins, ATF6α (670 amino acids) and ATF6β (703 amino acids). ATF6α is a potent transcriptional activator, while ATF6β has a low capacity to act as a transcription factor [56,57]. The ATF6α cytosolic domain contains basic leucine zipper (bZIP) DNA binding and a transcriptional activation domain, followed by a 20-amino acid transmembrane domain. Activation of ATF6a is a multistep process mostly studied during the UPR. Under ER stress, freed from GRP78, p90 ATF6α moves to the Golgi complex, where it is cleaved by site-1 proteases to form the p50 ATF6α that acts as a transcription factor in order to increase the level of resident folding factors [58]. 

During myogenesis, we show, firstly, that, for ATF6α to be activated, it requires p38 MAPK phosphorylation. Our results showing that the p38 MAPK inhibitor SB203580 impaired myogenesis suggest that this modification is essential to promote not only ATF6α activation but also differentiation of myoblasts. Indeed, in the presence of SB203580, besides the loss of ATF6α activation, p38 MAPK inhibition has the effect of downregulating either the expression of the marker of differentiation MyoG or of the downstream targets of ATF6α (GRP78/Bip; calnexin and calreticulin). As a consequence of this, the expansion of the ER membrane network is greatly reduced, suggesting that ATF6α is required during myogenesis in order to ensure an efficient quality control of secretory proteins in the multinucleated myotubes and the full development of sarcoendoplasmic reticulum.

Examining in deeper detail the molecular events underlying ATF6α activation, we show that phosphorylation by p38 MAPK of the threonine residues at position 166 of the ATF6α protein is essential for the proteolytic cleavage of p90 ATF6α at the Golgi complex and, as a consequence, for the import of p50 ATF6α inside the nucleus, where the truncated protein acts as transcriptional regulator. This event is very similar to that observed when inflammatory stimuli with IFN-γ-induce p38 MAPK activation, which in turn increases ATF6α phosphorylation and processing [23,44]. Similar to what we observe during myogenesis, ATF6α mutants in the p38 MAPK phosphorylation site (T166) fail to undergo proteolytic cleavage at the Golgi complex and nuclear import in IFN-γ-induced MEF cells [44]. Instead, ATF6α S130A, which is also a putative p38 MAPK phosphorylation site, maintains normal trafficking and processing features, suggesting that the S130 site is not essential for ATF6α activation.

In order to establish the function of ATF6α in the development of myotubes, we performed gene inactivation experiments, which showed that ablation of the *ATF6α* gene keeps C2C12 myoblasts in an un-differentiated state. This confirms that the ATF6α pathway is essential for muscle cell differentiation.

Further validation of this can been seen by the rescue experiments, in which these effects were reverted by the 3xFlag-ATF6α WT expression in the *ATF6α* KO cells, in which the ER-to-Golgi transport, cleavage and nuclear translocation of ATF6α regularly occurred and derived a differentiated phenotype. The treatment with SB203580, in the differentiation medium *ATF6α* KO cells transfected with 3xFlag-ATF6α WT vector led to a notable loss of differentiated morphology (Figure 5A), with the same biochemical pattern and cell distribution of ATF6α protein shown in un-transfected *ATF6α* KO cells.

On the other hand, p38 MAPK, through MyoD and MEF2, controls the expression of MyoG [59]. Although our evidence may indicate otherwise, we did not find, through bioinformatics analysis (JASPAR datasets), any regulatory sequence in the promoter region of MyoD or MyoG matching with known ATF6α binding sites [60,61]. Therefore, we exclude that myogenic factors expression could be under the direct control of ATF6α.

p38 MAPK may promote cell proliferation or induce cell cycle arrest, depending on the downstream pathways [23,62]. Intriguingly, emerging evidence suggests that ATF6α can influence cell cycle entry and can be implicated in the control of cell proliferation in several cells [63,64]. 

Moreover, it has been shown that ATF6α signaling may play a role in developmental apoptosis and differentiation programs during muscle development [27,65]. However, at present, the ways in which ATF6α participates in the development of the muscle cell is still unclear. 

Instead, we discovered, by genetic ablation experiments, that ATF6α contributes to cell cycle arrest, which is an essential prerequisite to start myoblasts differentiation. Therefore, since cells must exit the cell cycle, in order to differentiate a failure to exit the cell cycle would block all subsequent steps of differentiation. Therefore, it appears that the main mechanism through which ATF6 regulates myogenesis is the regulation of the exit from the cell cycle. This is supported by the finding that *ATF6α* knockout blocks the exit from the cell cycle as shown by the evidence that the gene ablation reduces the percentage of cells in the inactive G1 phase and increases the number of cells in the active S phase, thus preventing the exit from the cell cycle and therefore preventing differentiation. Moreover, the transfection with the ATF6α WT construct, in the *ATF6α* KO cells, was able to promote the exit of the cell cycle, reverting the effect of the KO of *ATF6α*. 

On the other hand, no effect was observed in the ATF6α KO transfected with the ATF6α T166A construct. A similar behavior was obtained after treatment with SB, the p38 MAPK inhibitor, in the *ATF6α* KO cells transfected with the ATF6α WT. Our findings demonstrate that ATF6 influences the cell cycle, favoring the exit from the cell cycle by promoting myoblast cell differentiation.

Some studies demonstrate that ATF6α can regulate differentiation by inducing a selective apoptosis to eliminate myoblasts, which are incompetent to differentiate [27,37,38,65]. This finding could be conflicting with ours. Nevertheless, although we cannot exclude this important function of ATF6α in our system we observed only low rates of apoptosis (Appendix A), which is possibly due to the low number of incompetent myoblasts and suggests that the proapoptotic function of ATF6α is irrelevant in our conditions. 

The malfunction of p38 MAPK has already been counted among the potential causes of muscle atrophy [66,67]. As regards this point, a significant effort has been made in order to identify natural or synthetic compounds, which could improve p38 MAPK activity to alleviate atrophy [68,69]. Muscle atrophy is also associated with ER stress, which activates the unfolded protein response (UPR) [16] but, while IRE1 and PERK appear to play a role in muscle regeneration after muscle tissue injury [16], the activation of ATF6α increases skeletal muscle adaptation [70], maintaining muscle tissue homeostasis [57]. 

In this paper we showed that treatment with DEX, a major inducer of muscle cell atrophy, inhibits differentiation of C2C12 myoblasts and reduces the amount of phosphorylated p38, thus impairing ATF6α activation and in turn myogenesis, which suggests that the p38 MAPK/ATF6α could also play a role in the regulation of atrophic events. 

## 5. Conclusions

Our work represents a further example of how the UPR can be activated to obtain specific outcomes in different cell types and tissues by using the same machinery. In our case, p38 MAPK activates the ATF6α pathway of UPR, but p38 MAPK alone is not sufficient to sustain myogenesis, since p38 MAPK phosphorylation is active even in the absence of ATF6α. Our results strongly suggest that ATF6α, throughout p38 MAPK phosphorylation, can ensure important changes to achieve the differentiated state during myogenesis. The main important step is the cell cycle arrest, which occurs by ATF6α. Further work is required to demonstrate that all the other effects we see are independent of this.

## Figures and Tables

**Figure 1 biomedicines-11-01457-f001:**
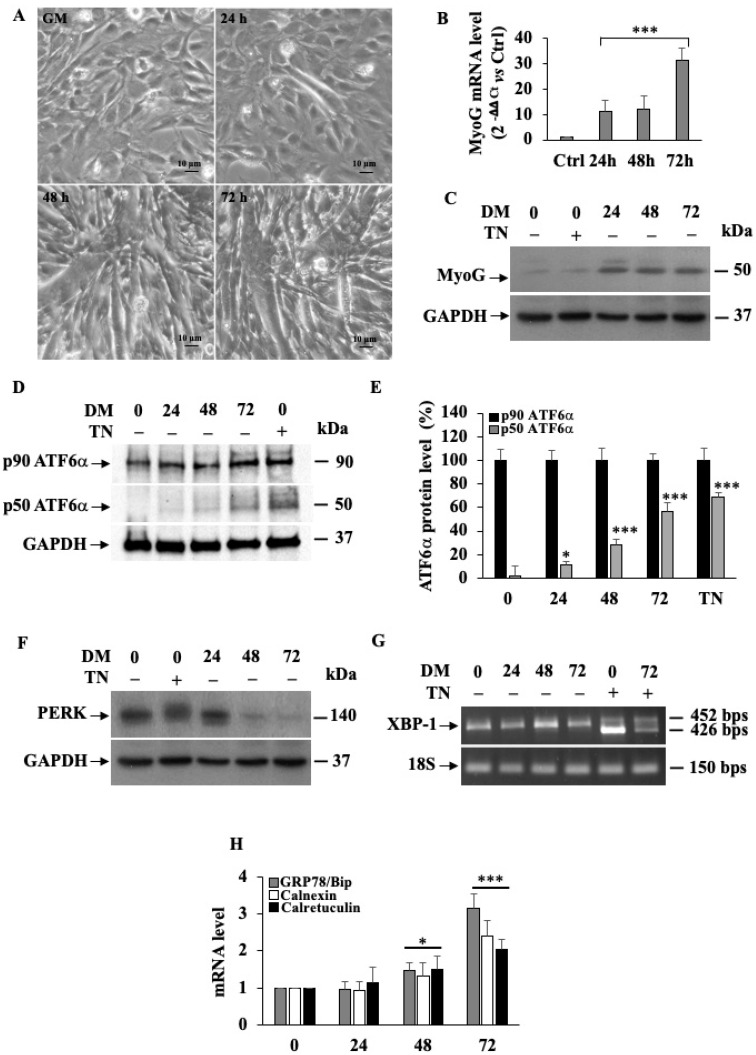
ATF6α activation in differentiating mouse C2C12 myoblasts. (**A**) Phase-contrast micrographs of C2C12 cultured in GM (point 0) or DM for 24, 48 and 72 h. Scale bar = 10 μm. (**B**) Quantitative qPCR analysis of MyoG mRNA expression levels in C2C12 cells. 18S was used as the internal control. *** indicates statistical significance compared to control cells (point 0) kept in GM (*p* ≤ 0.001). (**C**) Western blotting showing MyoG protein expression levels during the time course of differentiation. GAPDH was used as a loading control for cell lysates. (**D**) Western blot analysis of the ATF6α 90 kDa protein and its cleaved 50 kDa active form in C2C12 cells non-incubated or incubated with DM for 24, 48 and 72 h. GAPDH was used as a loading control for cell lysates. (**E**) Densitometric analysis of cleaved ATF6α and total protein expression levels. Experiments were performed in triplicate and quantitative results were performed as follows: p90 ATF6 alpha optical density/GAPDH optical density; p50 ATF6 alpha optical density/GAPDH optical density; and then p50 ATF6 protein were calculated as folds (%) respective to the p90 ATF6 values set as 100%. Each bar represents the mean ± SD (*n* = 3). * and *** indicate statistical significance compared to the controls (*p* ≤ 0.05; *p* ≤ 0.001, respectively). (**F**) Western blot analysis of PERK protein. GAPDH was used as a loading control for cell lysates. (**G**) RT-PCR was used to detect XBP1 mRNA forms. Total RNA fractions extracted from C2C12 cells incubated in GM (point 0) or in DM for 24, 48 and 72 h were analyzed. The migration on the gel of the un-spliced, 452 bp and spliced, 426 bp is indicated. (**H**) Quantitative qPCR analysis of ATF6α downstream target genes GRP78/Bip, Calnexin and Calreticulin mRNA expression levels in C2C12 cells. 18S was used as the internal control. * and *** indicate statistical significance compared to the controls (*p* ≤ 0.05; *p* ≤ 0.001, respectively).

**Figure 2 biomedicines-11-01457-f002:**
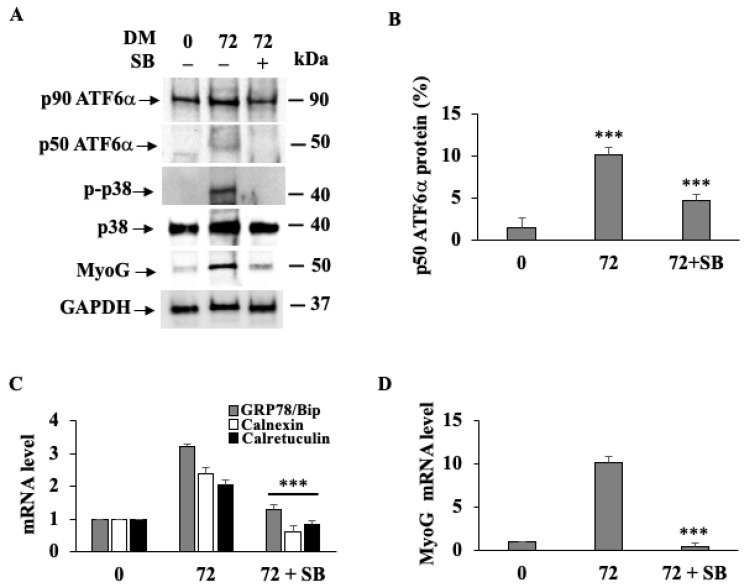
p38 MAPK triggers ATF6α activation during C2C12 myogenesis. (**A**) Western blots showing endogenous ATF6α protein, p-p38 MAPK and MyoG protein expression levels. GAPDH was used as a loading control of the cell lysates. (**B**) Densitometric analysis of cleaved ATF6α on experiments that were performed in triplicate. Each bar represents the mean ± SD (*n* = 3). *** indicates statistical significance compared to GM or DM (*p* ≤ 0.001). (**C**) Quantitative qPCR analysis of GRP78/BiP, Calnexin and Calreticulin. (**D**) MyoG mRNA expression levels in C2C12 cells. 18S was used as the internal control. *** indicates the statistical significance compared to DM = *p* ≤ 0.01.

**Figure 3 biomedicines-11-01457-f003:**
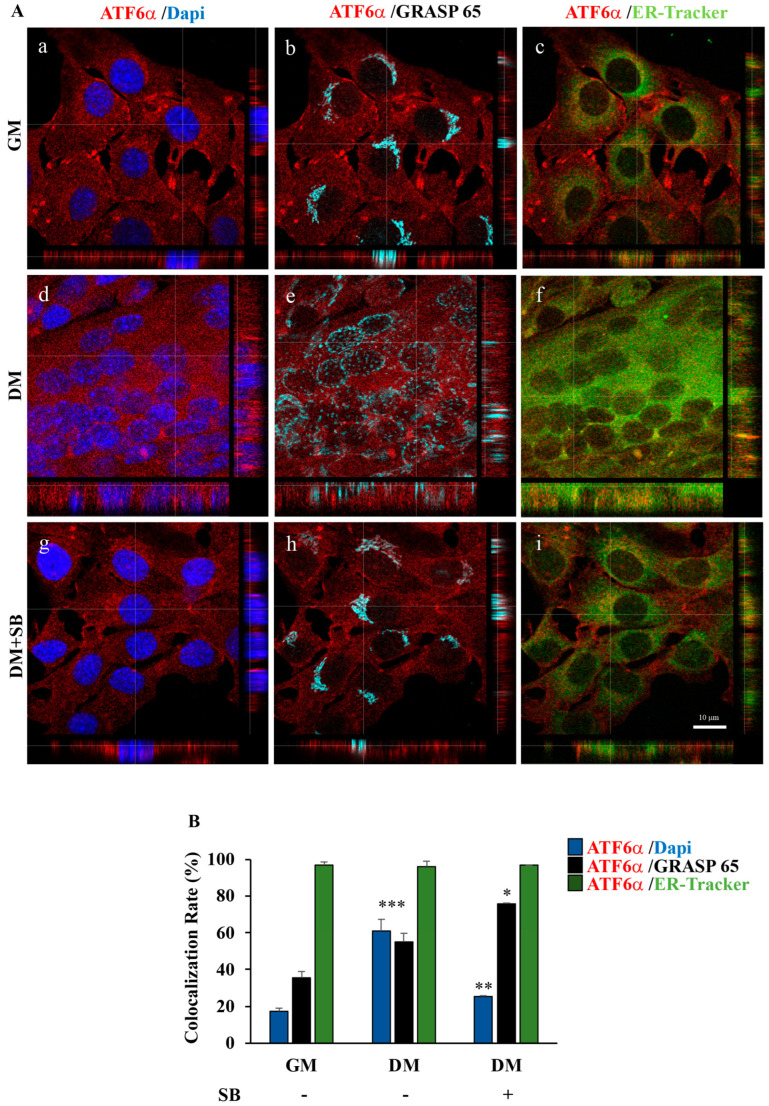
p38 MAPK is required for ATF6α nuclear import during C2C12 differentiation. (**A**) Representative images from a confocal z-stack with orthogonal side-views of C2C12 myoblast cells cultured in GM (panels **a**–**c**), DM for 72 h (panels **d**–**f**) and DM for 72 h treated with SB203580 (panels **g**–**i**). Cells were subjected to fluorescence analysis with ATF6α antibody (red), GRASP65 to visualize the Golgi (light blue) and in vivo endoplasmic reticulum-tracker (ER-tracker) (green). Nuclei were stained with DAPI (blue). Scale bar = 10 μm. Pictures were processed using LAS-AF Software to reconstruct the *z*-axis projection using stack images. (**B**) Quantitative analyses of the co-localization rate of endogenous ATF6α vs. nucleus, Golgi and ER, respectively, were measured by using the Leica SP5 software to quantify the number of co-localized pixels/total pixels in the image. The histograms represent the mean values of the % co-localization rate coefficient obtained in the various samples. To reduce intrinsic variability, we repeated this measurement on 50 cells for each experimental point. *, ** and *** indicate the statistical significance compared to GM or DM: * = *p* ≤ 0.5; ** = *p* ≤ 0.01; and *** = *p* ≤ 0.001.

**Figure 4 biomedicines-11-01457-f004:**
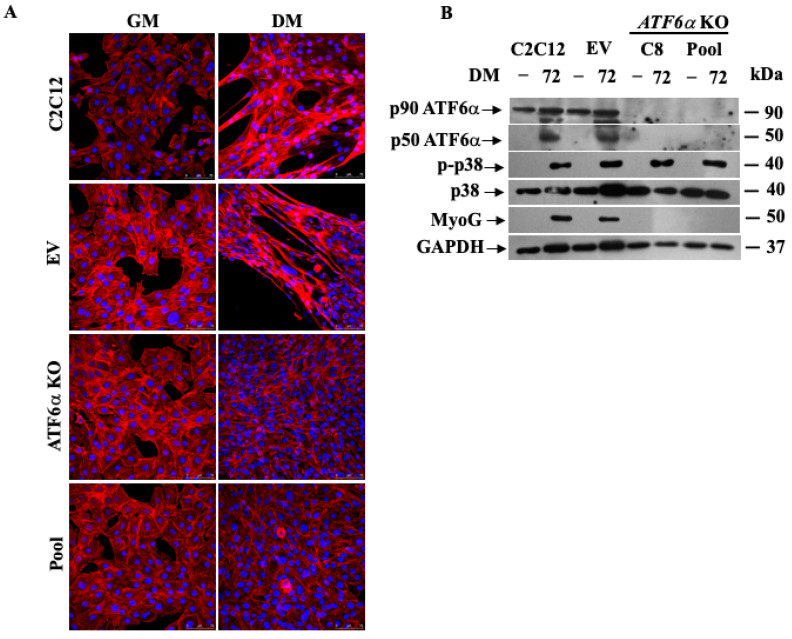
Effect of *ATF6α* KO on C2C12 myoblasts differentiation. (**A**) Cell morphology of C2C12 parental cell line, EV control cell clone, C8 selected clone and pool *ATF6α* KO cultured in GM or DM for 72 h; F actin was stained with phalloidin (red), and nuclei were stained with DAPI (blue). Scale bar = 75 μm. (**B**) Western blotting showing the protein expressions of endogenous ATF6α, p-p38 MAPK and MyoG in the C2C12 parental cell line, EV control cell clone or *ATF6α* KO cell clone and pool CRISPR/Cas9-induced *ATF6α* KO cell lines after 72 h of differentiation. GAPDH was used as a loading control of the cell lysates.

**Figure 5 biomedicines-11-01457-f005:**
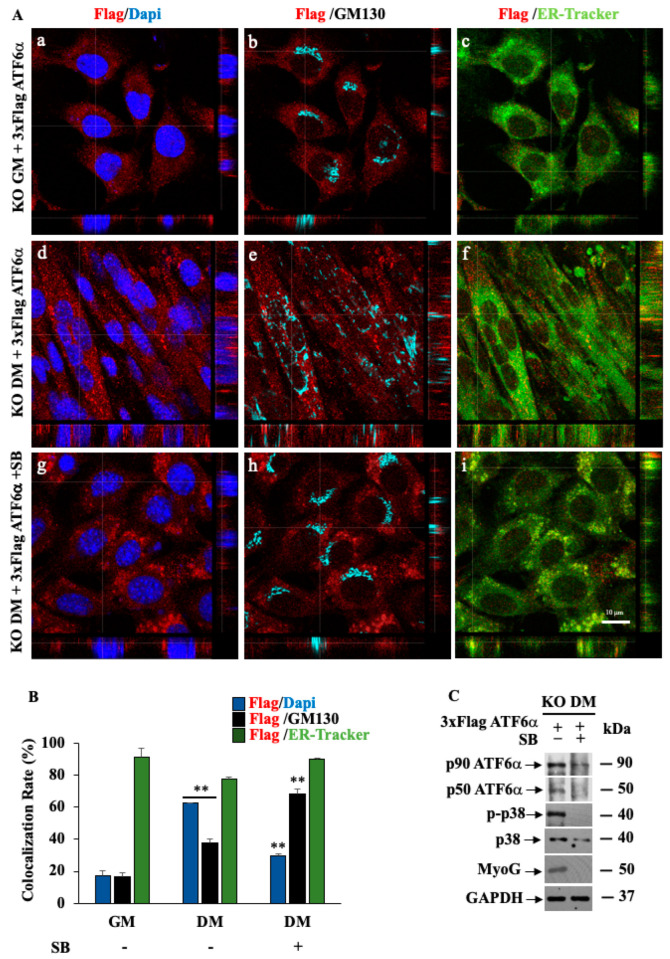
Genetic ablation of *ATF6α* impairs myogenesis of C2C12 myoblasts. (**A**) Representative images from a confocal z-stack with orthogonal side-views of KO *ATF6α* cells cultured in GM (**a**–**c**), DM for 72 h (**d**–**f**) and DM for 72 h, treated with SB203580 (**g**–**i**) and transfected with 3xFlag ATF6α WT. Cells were subjected to fluorescence analysis with the FLAG antibody (red), GM130 to visualize the Golgi (light blue) and in vivo endoplasmic reticulum-tracker (ER-tracker) (green). Nuclei were stained with DAPI (blue). Scale bar = 10 μm. Pictures were processed using LAS-AF Software to reconstruct the *z*-axis projection using stack images. (**B**) Quantitative analyses of the co-localization rate of exogenous ATF6α vs. nucleus, Golgi and ER were measured using the Leica SP5 software, quantifying the number of co-localized pixels in the image. To reduce intrinsic variability, we repeated this measurement on 50 cells for each experimental point. ** indicates the statistical significance of the differences between GM and DM: *p* ≤ 0.01. (**C**) Western blotting showing exogenous ATF6α protein, p-p38 MAPK and MyoG protein expression levels in KO *ATF6α* cells transfected with 3xFlagATF6 WT and treated or not with 10 μM of SB, inhibitor of p38, for 72 h of differentiation. GAPDH was used as a loading control of the cell lysates.

**Figure 6 biomedicines-11-01457-f006:**
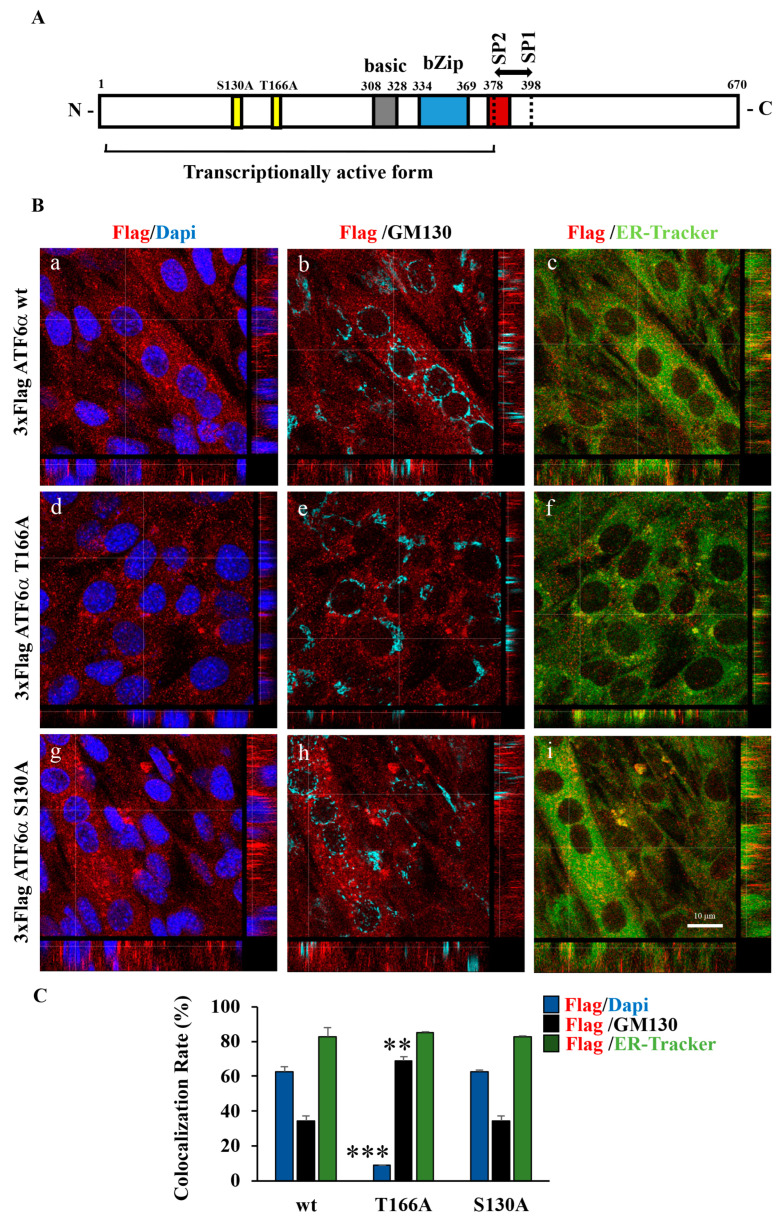
Mutation at the p38 MAPK phosphorylation site. T166 suppresses ATF6α nuclear import and the myogenesis of C2C12 myoblasts. (**A**) Topography of ATF6α protein structure in RE. Putative p38 MAPK phosphorylation sites T166 and S130 on ATF6α were identified by sequence analysis by using NetPhos 3.1 Server and indicated along with other important elements: N-terminal bZIP transcription factor domain; S1P and S2P proteolytic cleavage sites; and C-terminal domain. (**B**) Representative images from a confocal z-stack with orthogonal side-views of KO *ATF6α* cells cultured in DM for 72 h and transfected with 3xFlagATF6α WT (panels **a**–**c**), mutant constructs p3xFlagATF6 T166A (panels **d**–**f**) or S130A (panels **g**–**i**). Cells were subjected to fluorescence analysis with anti-Flag antibody (red), GM130 to visualize the Golgi (light blue) and in vivo endoplasmic reticulum-tracker (ER-tracker) (green). Nuclei were stained with DAPI (blue). Scale bar = 10 μm. Pictures were processed using LAS-AF Software to reconstruct the *z*-axis projection using stack images. (**C**) Quantitative analyses of the co-localization rate of exogenous ATF6α vs. nucleus, Golgi and ER were measured using the Leica SP5 software, quantifying the number of co-localized pixels in the image. To reduce intrinsic variability, we repeated this measurement on 50 cells for each experimental point. *** and ** indicate statistical differences with cells kept in DM and transfected with 3xFlag WT ATF6α (*p* ≤ 0.01 and *p* ≤ 0.001, respectively).

**Figure 7 biomedicines-11-01457-f007:**
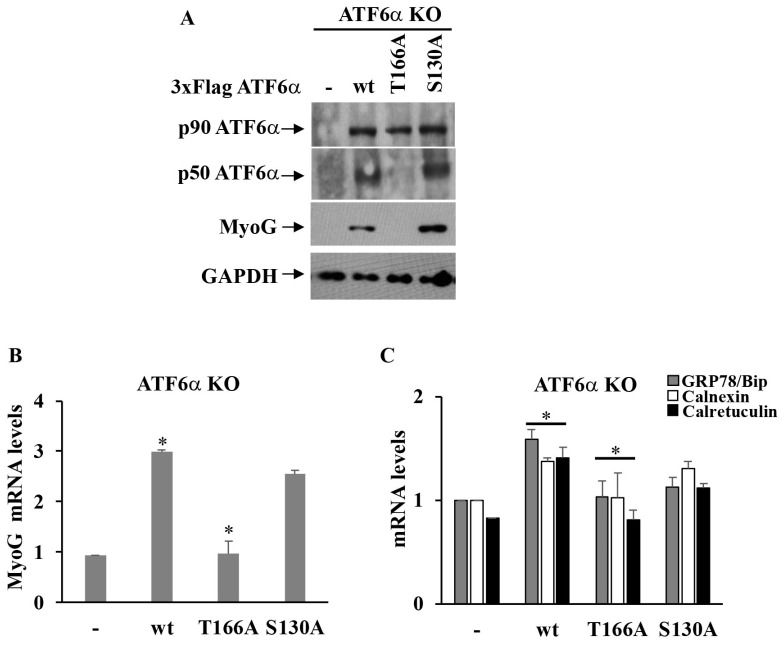
p38 MAPK phosphorylation site T166 of ATF6α regulates the differentiation of the C2C12 *ATF6α* KO cell clone. (**A**) Western blotting showing exogenous ATF6 protein, p-p38 MAPK and MyoG protein expression levels in the EV control cell clone and KO *ATF6α* cells kept in DM for 72 h. Cells were transfected with p3xFlagATF6α WT and mutant T166A or S130A p3xFlagATF6α constructs. GAPDH was used as a loading control of the cell lysates. (**B**,**C**) Quantitative qPCR analysis of GRP78/BiP, Calnexin and Calreticulin and MyoG mRNA expression levels in the EV control cells and ATF6α KO cells. 18S was used as the internal control. * Indicate *p* ≤ 0.05 value.

**Figure 8 biomedicines-11-01457-f008:**
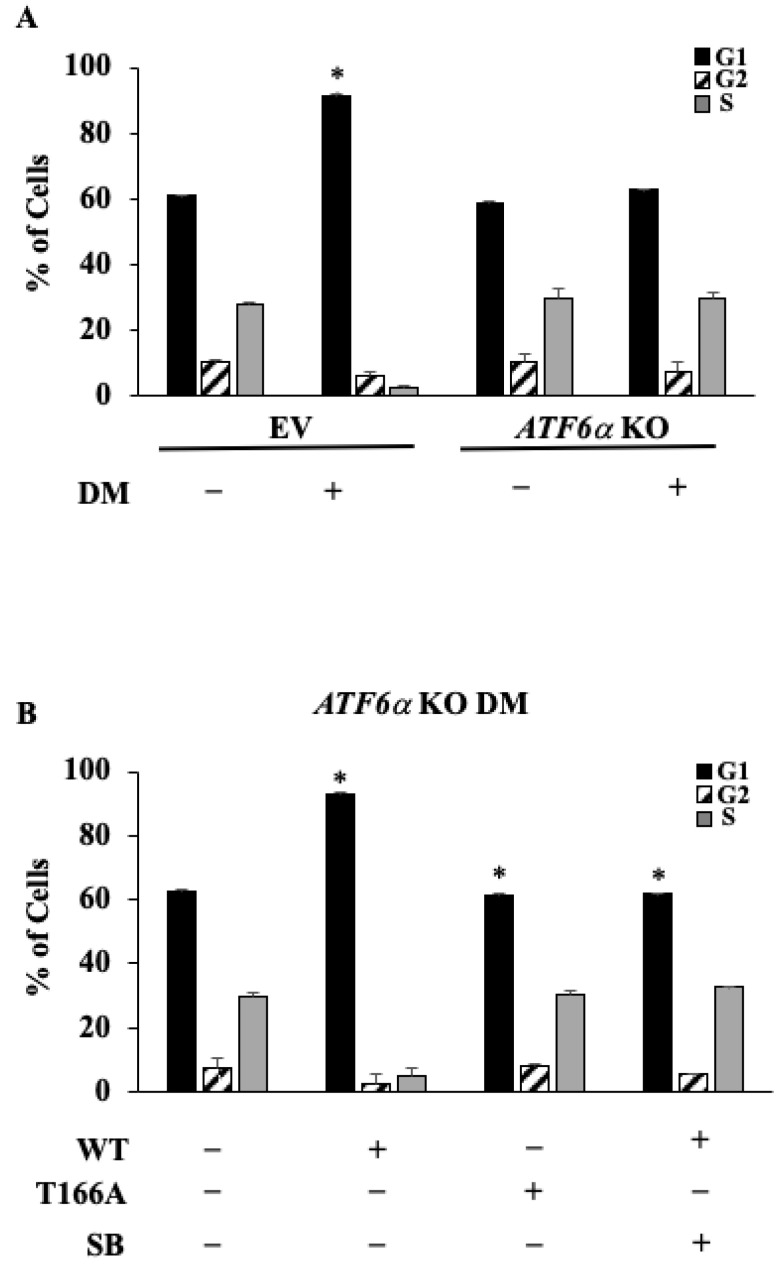
Effect of *ATF6α* KO on the cell cycle control during myoblast differentiation. (**A**) The control C2C12 (EV) and *ATF6α* KO cells incubated in DM or kept in GM for 72 h. (**B**) The WT and mutant T166A ATF6α constructs were transfected into the *ATF6α* KO cells. Cells were kept in DM or with SB203580 as indicated. Flow cytometry analysis of the cell cycle was evaluated after the indicated incubation time and reported as a percentage of cells found in each cell cycle phase. Data from triplicate experiments are reported as *, which indicates *p* values ≤ 0.05.

**Figure 9 biomedicines-11-01457-f009:**
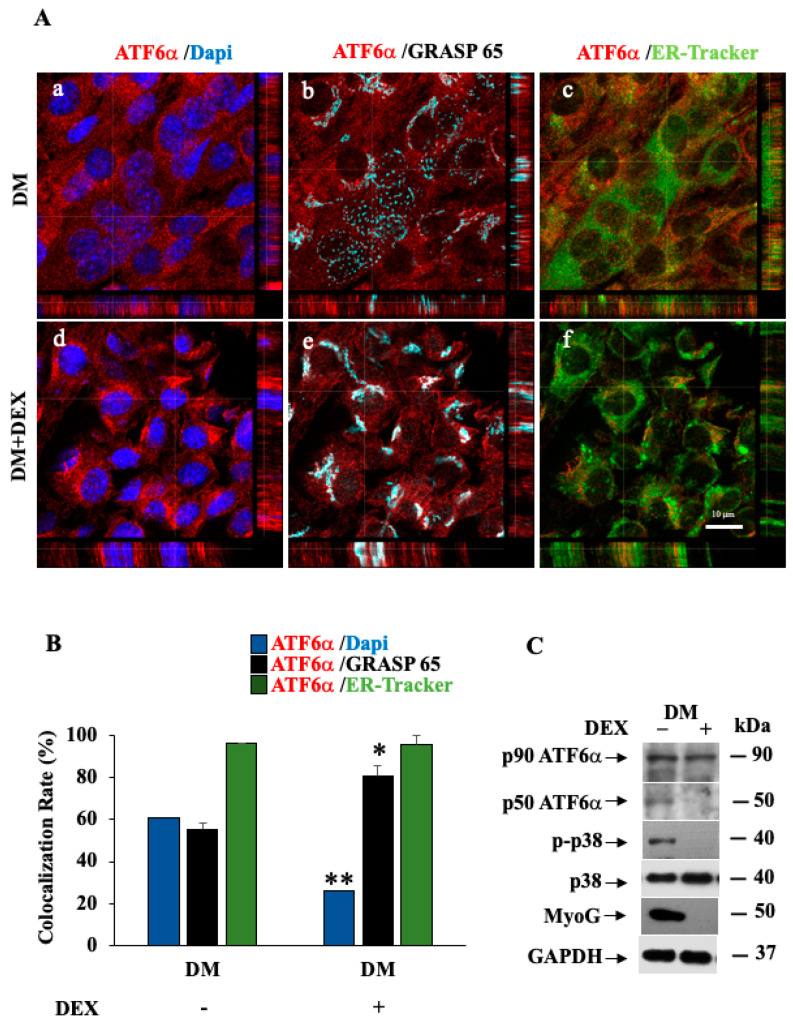
p38 MAPK/ATF6α pathway is impaired in DEX-induced myotube atrophy. (**A**) Representative images from a confocal z-stack with orthogonal side-views of the C2C12 myoblast cells cultured in DM (panels **a**–**c**) for 72 h, in the absence or presence with 10 μM DEX (panels **d**–**f**). Cells were subjected to a fluorescence analysis with ATF6α antibody (red), GRASP65 to visualize the Golgi (light blue) and in vivo endoplasmic reticulum-tracker (ER-tracker) (green). Nuclei were stained with DAPI (blue). Scale bar = 10 μm. Pictures were processed using LAS-AF Software to reconstruct the *z*-axis projection using stack images. (**B**) Quantitative analyses of the co-localization rate of endogenous ATF6α vs. nucleus, Golgi and ER were measured using the Leica SP5 software, quantifying the number of co-localized pixels in the image. To reduce intrinsic variability, we repeated this measurement on 50 cells for each experimental point. * and ** indicate the statistical significance of differences obtained from GM- or DM-treated cells (*p* ≤ 0.5 and *p* ≤ 0.01, respectively). (**C**) Western blotting showing endogenous ATF6α protein, p-p38 MAPK and MyoG protein expression levels in C2C12 cells. Cells were subjected to treatment with DEX for the in vitro atrophy model for 72 h. GAPDH was used as a loading control of the cell lysates.

## Data Availability

Not applicable.

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
