# Peer review of "Myogenesis in C2C12 Cells Requires Phosphorylation of ATF6α by p38 MAPK"

_biomedicines, 2023, doi:10.3390/biomedicines11051457_

Round 1

Reviewer 1 Report

Please see attached pdf

Author Response

Answers to Reviewer 1

Reply to the overall comments:

We thank the reviewer for the great effort that has put into pointing out the weaknesses and inaccuracies present in the work and for giving us helpful suggestions. In particular, we have found very fair the criticism concerning the term atrophy, which we have improperly used. More importantly, we thank the reviewer for   the observations regarding ATF6a, whose expression is required to promote the exit from the cell cycle that in turns favors differentiation. We tried to clarify this two main aspects and we hope that doing that we improved the quality of our work. Here is a point by point list of the answers to the reviewer suggestions and criticisms. 

Abstract:

Line21: “participate to” was changed to “participate

Line 26: “getting the inability of C2C12 myoblast” sentence was rewritten as suggested.

Line 31: “induced atrophy by the use of dexamethasone” was changed to “inhibited differentiation by using dexamethasone”.

Line 32: was modified accordingly

Introduction:

Line 42: “consisting in” was changed to “consisting of”

Line 48: comma was removed

Line 63: we “also” and changed from “participate to” to “participate in”

Line 80: we changed from “contents” to “contexts”

Line 85 “mouse myoblasts C2C12” was changed to “the mouse myoblast cell line C2C12”

Line 86/87: according to the reviewer suggestion the sentence was deleted.

Materials and methods:

Line 90: we specified from which source we obtained and we inserted a reference of a previous work where we used the same C2C12 cells.

Line 98: we thank the reviewer for this important observation. We specified in the text that cells kept for 24 h in DM were treated with dexamethasone (DEX) for 48 h. Since 24 h in DM are not sufficient to obtain myotubes we used the term differentiation inhibition instead of atrophy throughout the text

Line 112: “western blotting analysis C” was deleted.

Line164: we understand the observation. However, the ribosomal RNA small subunit (18S) has been successfully used as reference gene in important papers such as https://rdcu.be/c8Km0, in which the authors used 18S to normalise the expression of several muscle tissue specific genes in differentiating C2C12 cells (see also: https://doi.org/10.1371/journal.pone.0020780. In addition, our findings showing that SB203580 suppresses MyoG mRNA expression (p ≤ 0.001) shown in Figure 2D are in line with the results obtained by Rampalli, S et al. 2007, making us more confident of having chosen a reliable reference gene.

Line167-173: We apologize for missing the description of how we designed primers. We specified in the paragraph 2.5 the complete description.

Line 208: “PI” stays for Propidium Iodide and we have indicated the full name in the materials and methods.

Line 257: we thank the reviewer for the observation accordingly we deleted the sentence regarding ATF4

Line 275: the sentence was re-written accordingly.

Line 282: cell images showing the effect of SB203580 are displayed in Figure 3A

Figure 2B: we have modified the figure legend as suggested by the reviewer.

Figure 2D: figure legend was modified as suggested.

Line 309: “contest” was changed to “context”

Figures 3B, 5B and 6C (Colocalization Rate %): we have sorry for the lack of clarity; we have reported a more detail description of the methods in M&M paragraph 2.6.

Lines 355-361; Line 377 and Line 380: “the text does not agree with what the labels on figure 5 suggest” Labels of figure 5 a and c were modified indicating that experiments were not done in KO ATF6a cells but in KO ATF6a cells transfected with plasmid vector expressing 3xFLAG ATF6a wild type as also previously indicated in the results section and in the figure legend.

Line 440: the full name of PI was indicated

Line 463: we have modified the text as suggested as suggested by the reviewer.

Line 464: we clarified in the text that DEX inhibited differentiation.

Line 467: see above and answer to Line 98
Line 474: we specified in the text that myogenin expression, p38MAPK and ATF6α activity decreased because DEX has inhibited differentiation

Discussion

Line 494: as suggested by the reviewer the discussion was redone and made more focused around the observation he made.

Line 494/line 513: According to the reviewer suggestion, we have modified inconsistencies between the use of myotubules and myotubes.

Line 553-556: we modified the text according the observation that the effects of ATF6α relate more to cell cycle exit, which would come before the induction of myogenin

Line 558: we have tried to clarify these concepts better in the discussion.

Conclusions:

Line 577:  the was removed by the sentence

Line 578: In agreement with the reviewer, we have modified as suggested

Line 579-581:  as we already stated in the reply to overall comments we thank the reviewer for the observations and we tried to clarify this main aspect in the conclusion

Reviewer 2 Report

Results

Fig.1

A. The photos are too small, the central nuclei in the myotubes are not visible.

D. Western blot (WB). Column 4 (DM 72h), on the lane from p50 ATF alpha, a white spot is observed caused by an air bubble between the gel and the nitrocellulose membrane during transfer, as a result the read optical density is incorrect.

Fig. 2

A. The GAPDH bands are unequal. However, it can be seen that the fragments of nitrocellulose membranes that appear in the images do not come from the same membrane, so it is not known from which transfer the GAPDH bands come. There is no evidence, not even in the supplementary material provided, for protein loading control for all membranes, respectively for all proteins tested.

The observation is also valid for the other figures that present the WB data, even if it only concerns the presence/absence of a specific protein. The loading control (GAPDH) for all membranes should be seen in the supplementary material.

B. It is not specified that a regularization was made of the investigated proteins (for example p90 ATF6 alpha or p50 ATF6 alpha) according to GAPDH (protein optical density/GAPDH optical density) in order to correctly calculate the optical density of these proteins. When the loading control is not equal for all samples, this regularization must be done.

It is necessary for the Western blot analyzes to be redone and an additional supplementary material to be presented for each entire nitrocellulose membrane, with the proteins of interest identified and with the corresponding loading control (GAPDH).

Discussion

The study is a complex one, with many experimental in vitro models that give rise to an important number of results. But the explanation and interpretation of these results are poor. The impression left by the authors is that some of them actually worked, and others tried to discuss the results, but without having an overview or the necessary knowledge to analyze them in depth and interpret them correctly. Even the English language in Discussion chapter is deficient, with errors and mistakes of expression, as if the authors were not familiar with specialized terms, such as myotubes (named myotubules that appear in myoblasts, and not myotubes that differentiate from myoblasts; or called multinucleated tubules).

Some passages from the Discussion should have been placed in the Introduction, such as the description of the activity of PERK, IRE1. The analysis of the original results regarding ATF6 alpha should be much more extensive.

A correction of the English language and a serious reverification of the scientific expression would be necessary, especially in Discussion chapter. It would also be useful for the discussions to be more anchored on the results, to analyze them more extensively, with rigorous explanations and in accordance with the data from the literature.

Author Response

Results

A:The photos are too small…

Reply: we enlarged phase-contrast images in which the formation of myotubes can be clearly observed. Phase-contrast does not allow to see nuclei, which can be seen in the subsequent experiments performed by confocal microscopy.

  1. Western blot (WB)…Column 4 (DM 72h), on the lane from p50 ATF alpha, a white spot ….

Reply: According to the reviewer WB were redone and a new figure Fig. 2 has been inserted

Fig 2

  1. The GAPDH bands are unequal…..

Reply: as we previously stated we repeated the experiment, prepared a new figure and add the original images in the supplementary file. Densitometric analysis was done on at least 3 experiments and therefore we are confident of the data obtained. Unfortunately, the quality of commercially available ATF6 antibodies is not the best. For this reason, we validated our data also with the exogenous ATF6 in rescue experiments. Original images are shown in the supplementary file.

  1. It is not specified that a regularization was made of the investigated proteins according to GAPDH….

Reply: We understand the reviewer concerns. All WBs that are shown in the work come from the same membrane as demonstrated in the original images file. We also would like to point out that to optimize laboratory time and resources, in some cases we have cut filters and therefore images have different backgrounds. We included the densitometric analysis of all WBs in a new supplementary figure, whose results are in line with immunofluorescence and qPCR results. How densitometric analysis of p90 ATF6 alpha and p50 ATF6 alpha was carried out is described in the Figure legend

Discussion

Discussion was completely rewritten.

Discrepancies in terms such as myotubes, myotybules etc… were corrected.

Description of the activity of PERK, IRE1 was moved to the introduction paragraph

The English language was corrected by native English speaker

Round 2

Author Response

Replies to reviewer 1

We thank the reviewer for the fruitful suggestions and the necessary corrections that he has indicated.

Here are our point by point answers to the second round of reviews

Abstract:

As suggested we have added a sentence at the end of the abstract.

Results:

Line 509-510 was modified as suggested

Line 529 was rewritten as indicated

Line 531-532: the sentence was so deleted most of it as requested

Discussion and Conclusions

Previous comments about the effects on exiting the cell cycle were crucial and gave the work much more meaning. We thank the reviewer above all for this.

Author Response

Replies to reviewer 2.

We thank the reviewer for the careful and meticulous review work.

We have done our best to respond adequately to the requests of this second round. In particular, experiments WBs shown in Fig. 1D and E and in Fig. 2 were repeated and re-quantized. We feel that the revised version of the paper has gained more in quality and may attract more interest from the reader.

We made the necessary corrections and here are our point by point answers to the second round of reviews

Rev: Line 174..

Replay: Line 174 was reformulated. "determined by the Bradford… " and "filter" was corrected with "membrane" (nitrocellulose membrane).

Rev: The original WB images, supplemented, should also contain the molecular mass marker….

Reply: Molecular original image file was modified by adding to each film the relative position of MW marker used, which was specified in Materials paragraph 2.4. In the images obtained by the Biorad Chemidoc acquisition System MW markers are visible both in color and in black and white.

Rev: I don't agree with the images that look like a drawing on a canvas, in Fig 2. of the article….,

Reply: Experiment shown in Fig. 2 has been redone and also in this case original images are shown in the original image file (OI).

Rev: In the supplementary material for WB, …. the background from GAPDH ..MyoG and p-p38… …. ATF6 alpha and p38.

Reply: As previously mentioned we repeated the experiment, prepared a new figure and added the original images in the OI file.

Rev: … I would have liked to see the fragments cut from the same membrane grouped, at the same dimensions, with the GADPH of the respective membrane, in order to have the certainty of identical loading of the lysates in the gels.

Reply: To answer the reviewer request we repeated WBs as shown in Figures 1D and 2A. The OI files uploaded shows the image acquisition performed by the Biorad Chemidoc System, which results in one .scn file for the marker and on .scn file for the detected signals of the protein of interest. Thus, we generated figures showing both the marker and Western Blot signals from the same membrane shown in the original images file. We also added the Ponceau-S in original images file. The other Western blotting experiments were acquired with plates. We agree with the referee that this system makes it difficult to demonstrate that the result of the experiment comes the same membrane. However, to distinguish among proteins having similar molecular weights, we stripped the membrane before proceeding with the subsequent incubations of the antibodies and this could explain background differences.

As for the ATF6 antibody, few companies produce this reagent and few ATF6 antibodies recognize both p90 and p50 forms. Initially, we used an antibody that is no longer in the market (Imgenex), which worked very well for westerns but not for immunofluorescence. Instead, the Santa Cruz ATF6 antibody is not the best but works also for immunofluorescence experiments

Rev: In Fig.1, image D (article), the bands from ATF6 alpha p90 and p50 in the differentiation medium at 72 hours and after treatment with tunicamycin are almost identical, which is not evident in the graphic representation in E.

Moreover, why is not made the graphical representation of the optical densities, but another calculation is described. Even the English expression of this calculation is not correct, when explaining figure 1 E. Why p90 ATF6 values are set as 100%? Is the reasoning correct? Explain.

Reply: the densitometric analysis was done on at least 3 experiments data are statistically significant. Instead, the image shows only one representative image. However, to answer the reviewer we repeated the experiment and prepared a new figure, adding the original images in the OI file.

Densitometric analysis of p90 ATF6 alpha and p50 ATF6 alpha was carried out as follows:

p90 ATF6 alpha optical density/GAPDH optical density; p50 ATF6 alpha optical density/GAPDH optical density.

The values were obtained as follows:

p90 ATF6 alpha : 100 = p50 ATF6 alpha : X

X= (p50 ATF6 alpha/p90 ATF6 alpha) * 100

The p50 ATF6 alpha to p90 ATF6 alpha ratio is used to normalize the cleaved protein respect to the total protein. Conventionally, the same normalization is used for phosphorylated proteins respect to the total protein. Then we have chosen to express the value as a percentage. Moreover, the densitometric analysis shown in the figure was done on 3 experiments and statistical significance is reported in the figure legend.

The graph shown in the Supplementary Figure 1 (Figure Supp. 1D) indicates the mean optical density values of p90 ATF6 alpha and p50 ATF6 alpha, also including the last experiment shown in figure 1D, normalized with the GAPDH, without p50 ATF6 alpha to p90 ATF6 alpha ratio. The results are comparable to those obtained by referring to ATF6 p90 as 100%.

Rev: In the Results it is said that in Fig. 1D it is represented "activation of the endogenous ATF6α assessed by measuring the amount of the un-cleaved p90-ATF6α protein against the cleaved p50 form of ATF6α in C2C12 myoblast, cultured in GM or in DM for the times indicated (Figure 1D)". It is not true, in this figure it can be seen only what is happening in the DM culture medium.

Reply: Conventionally, in the experiment shown in Fig. 1D the DM time 0 corresponds to the cells incubated in GM only. This has been clearly written in the figure legend.

Rev:…for moments 0 and 24 the bands for both forms of ATF6 alpha are not clearly visible and cannot be quantified correctly.. image in question, very blurry, does not explain the attached optical density graphs. Fig. 1 D and E are not a suitable result for publication.

Reply: As above described, we repeated the experiment and prepared a new figure 1D and E, the original images are in the OI file.